# *Toxoplasma gondii* chronic infection decreases visceral nociception through peripheral opioid receptor signaling

Alexis Audibert[1,2], Xavier Mas-Orea[2], Léa Rey[2], Marcy Belloy[1], Emilie Bassot[1], Louise Battut[2], Gilles Marodon[3], Frederick Masson[1], Matteo Serino[2], Nicolas Cenac[2], Gilles Dietrich[2], Chrystelle Bonnart[2*], Nicolas Blanchard[1*]

**1** Toulouse Institute for Infectious and Inflammatory Diseases, Infinity, Inserm, CNRS, University of Toulouse, Toulouse, France, **2** IRSD, Inserm, INRAE, ENVT, University of Toulouse, Toulouse, France, **3** Centre d'Immunologie et des Maladies Infectieuses (CIMI-PARIS), Inserm, CNRS, Sorbonne University, Paris, France

☙ These authors contributed equally to this work.
* nicolas.blanchard@inserm.fr (NB); chrystelle.bonnart@inserm.fr (CB)

## Abstract

By eliciting immune activation in the digestive tract, intestinal pathogens may perturb gut homeostasis. Some gastrointestinal infections can indeed increase the risk of developing post-infectious irritable bowel syndrome (PI-IBS). Intriguingly, the prevalent foodborne parasite *Toxoplasma gondii* has not been linked to the development of PI-IBS and the impact of this infection on colon homeostasis remains ill-defined. We show in a mouse model that latent *T. gondii* decreases visceral nociceptive responses in an opioid signaling-dependent manner. Despite the accumulation of Th1 and cytotoxic T cells in the colon of latently infected mice, the selective invalidation of enkephalin gene in T cells ruled out the involvement of T cell-derived enkephalins in hypoalgesia. These findings provide clues about how this widespread infection durably shapes the gut immune landscape and modifies intestinal physiological parameters. They suggest that in contrast to other gut microbes, *T. gondii* infection could be negatively associated with abdominal pain.

## Author summary

*Toxoplasma gondii* is a common parasite that infects many people worldwide, often without causing noticeable symptoms. Gut infections can lead to long-term digestive issues, like post-infectious irritable bowel syndrome (PI-IBS). Although *T. gondii* is a gut infection agent, it has not been linked to PI-IBS so far. In this study performed in an experimental mouse model, we examined how *T. gondii* latent infection affects the gut. We unexpectedly discovered that chronically infected mice experience less gut pain compared to uninfected animals. This

**Data availability statement:** Colon mucosa-associated microbiota sequences are available on the Sequence Read Archive (SRA) database with the assigned identifier PRJNA1224722. https://www.ncbi.nlm.nih.gov/bioproject/PRJNA1224722 All other relevant data are within the manuscript and its Supporting Information files.

**Funding:** This work was supported by institutional grants from Inserm, PIA PARAFRAP Consortium (ANR-11-LABX0024 to NB), "Agence Nationale pour la Recherche" (ANR-19-CE15-0008 TRANSMIT to FM/NB; ANR-22-CE14-0053 NINTENDO to NB ; ANR-22-CE15-0018 ImmunUP to NB), "Fondation pour la Recherche sur le Cerveau" AAP2021 to NB. AA was supported by a doctoral fellowship from the French Minister of Research and a 4th year PhD fellowship from "Fondation pour la Recherche Médicale" (FRM FDT202304016672). The funders had no role in study design, data collection and analysis, decision to publish, or preparation of the manuscript.

**Competing interests:** The authors have declared that no competing interests exist.

effect is linked to the body's natural pain-relief system, involving opioid molecules. This infection also led to long-lasting changes in immune cells of the gut, including T lymphocytes. Since T cells can produce opioids, we suspected them to play a role in the reduced pain phenotype. However, mice lacking a key opioid in their T lymphocytes still showed the same pain decrease, excluding this possible mechanism. In summary, these findings suggest that, unlike other gut infections, latent *T. gondii* infection may have a pain-relieving rather than a pain-inducing effect. They help better understand how infections influence gut health in unexpected ways, and in the future, they might contribute to decipher new pain control mechanisms.

## Introduction

Most often, infections are rapidly resolved, allowing the organism to return to its initial status. However, in some cases, infections can have long-lasting consequences, thereby negatively or positively regulating the development of chronic diseases [1–3]. As examples, a mouse model suggested that early-life brain viral infection can predispose to the development of brain autoimmune disease during adulthood [4] and in humans, Epstein-Barr virus infection increases by over 30-fold the risk to develop multiple sclerosis [5].

As one of the largest interface between the host and the environment, the gastrointestinal (GI) tract is highly exposed to pathogens, which can profoundly shape the mucosal immune environment [6]. The GI tract is composed of different cellular layers. Below the epithelium, which surface is exposed to the gut microbiota, lie (i) a lamina propria containing fibroblasts and immune cells, (ii) muscular layers, and (iii) a neuronal network divided into an intrinsic (enteric nervous system, ENS) and extrinsic peripheral neuronal compartment. These entities exert specific functions that are crucial for tissue homeostasis. The ENS regulates intestinal functions such as gut motility, secretion, permeability, and blood flow. Extrinsic neurons relay peripheral information, such as visceral pain, to the central nervous system (CNS) through afferent projections from neurons (nociceptors) that have their soma in the dorsal root ganglia (DRG). Visceral pain transmitted by nociceptors results from the integration of excitatory (pro-nociceptive) and inhibitory (anti-nociceptive) signals present within the surrounding environment. Pro-nociceptive mediators like pro-inflammatory cytokines or some proteases [7], increase nociceptor firing, leading to visceral hypersensitivity, while anti-nociceptive mediators such as endogenous opioids, decrease nociceptor activity, thus relieving visceral pain [8–11]. As drivers of inflammatory mediators, several intestinal pathogens causing infectious gastroenteritis enhance pro-nociceptive signals and increase the risk of developing Irritable Bowel Syndrome (IBS), a phenomenon known as Post-Infectious IBS (PI-IBS) [12]. According to Rome IV criteria, IBS is a functional chronic disorder characterized by changes in bowel habits and abdominal pain without obvious gut inflammation or macroscopic lesions [13]. In humans, PI-IBS has been reported following infections

with pathogens such as bacteria (*Escherichia coli*, *Salmonella*, *Campylobacter jejuni*), parasites (*Giardia intestinalis* [14], *Cryptosporidium* [15]) and viruses (*Norovirus* [16]), suggesting that common immune-mediated mechanisms possibly trigger the development of this syndrome [17]. Interestingly, one study suggested that visceral hypersensitivity in PI-IBS could arise from prolonged nociceptor activation and sensitization due to incomplete resolution of immune responses, thus leading to persistent abdominal pain despite apparent recovery from acute gastroenteritis [18]. Moreover, *Blastocystis*-infected rats exhibit a PI-IBS-like phenotype characterized by a non-inflammatory visceral hypersensitivity [19] and *Citrobacter rodentium* infection of mice evokes hyperexcitability of colonic DRG neurons which persists following resolution of the infection [20], indicating that IBS features can be recapitulated in animal models of infection. Intriguingly, there is currently no epidemiological evidence that the highly prevalent foodborne parasite *Toxoplasma gondii (T. gondii)* is linked to the development of PI-IBS.

*T. gondii* is an obligate intracellular protozoan parasite able to infect any warm-blooded animal, including humans. It is estimated that around 30% of the world population, reaching up to 50% in some areas, has been exposed to this parasite [21]. Infection occurs after ingestion of contaminated food or water, allowing parasite cysts or oocysts to penetrate the gut and invade the host through the small intestine. Parasites then proliferate as fast-replicating tachyzoites, causing tissue inflammation, bacterial translocation, and strong type 1 immune responses including durable Th1 CD4+ T cell responses [6,22,23]. During this initial acute phase of infection, tachyzoites disseminate systemically and reach the CNS, where chronic infection is established. The long-term chronic phase is characterized by clearance of the parasite from the periphery (including the gut) and persistence of the parasite in the CNS, retina and skeletal muscles in the form of slow-replicating bradyzoïtes located within intracellular cysts [24,25]. While the impact of *T. gondii* chronic infection on brain function has been extensively studied [26,27], much less is known about the potential long-term consequences of *T. gondii* infection on gut homeostasis. One pioneer study has reported that by inducing bacterial translocation during the acute phase, *T. gondii* infection promotes the development of microbiota-specific CD4+ memory T cells with a pro-inflammatory Th1 phenotype. Interestingly, these T cells can persist in the gut for more than 200 days post-infection [23]. While microbiota dysbiosis is well-documented during the acute phase of infection, current literature suggests that microbial alterations are minimal or absent in the chronic stage of *T. gondii* infection [28–31]. Regardless, it was recently shown that chronic *T. gondii* infection exacerbates the severity of intestinal damage caused by chemical injury, due to heightened activation status of monocytes in chronically infected mice [28]. In rats, chronic *T. gondii* infection has been reported to cause damage of the colonic mucosa and to induce death of some neurons of the submucosal or myenteric ENS plexi, without altering the gastrointestinal transit time or the fecal pellet output [32,33].

Together, these data suggest that as other intestinal pathogens, by shaping the neuroimmune environment of the gut, *T. gondii* infection could promote the development of PI-IBS. To address this question, we explored the long-term consequences of latent *T. gondii* infection on the colonic microenvironment and on visceral nociceptive responses. Contrary to our expectations, we found that chronic *T. gondii* infection decreases visceral nociception in an opioid receptor signaling-dependent manner, in the absence of macroscopic colonic inflammation and with only a modest change in the colon mucosa microbiota. This effect was associated with a long-lasting increase in colonic T cells displaying a Th1 or cytotoxic phenotype, but could not be explained by T cell-produced enkephalins. These findings shed light on the long-term impact of this widespread foodborne infection on the gut neuroimmune landscape and physiology.

## Results

### *Toxoplasma gondii* chronic latent infection decreases visceral nociceptive responses

As C57BL/6 mice are naturally susceptible to *T. gondii* infection and develop encephalitis following infection with a type II strain, we used an established model of infection of C57BL/6 mice with a modified type II strain expressing the immuno-dominant GRA6-OVA model antigen (Pru.GFP.GRA6-OVA), which elicits a strong CD8+ T cell response able to effectively control the parasite burden during chronic stage [34], thereby mimicking the pathophysiology of latent human infection. To examine the importance of the primary intestinal phase of infection on the pathophysiological outcome, mice were infected

either *per os* by administration of 10 cysts or intraperitoneally (ip) by injection of 200 tachyzoites (Fig 1A). As previously reported [35], *T. gondii* infection induced a transient weight loss during the acute phase of infection (Fig 1B), whatever the mode of parasite administration. At 10 weeks post-infection (pi), none of the *per os*-infected mice exhibited detectable parasite burden in the colon while some ip-infected mice still had detectable parasites in the colon. Both models displayed similar parasite loads in the brain (Fig 1C).

To assess visceral sensitivity in non-infected (ni) *versus* chronically infected mice (*per os* or ip, 10 weeks post-infection), we measured the visceromotor response (VMR) elicited by colorectal distension. In this assay, VMR measurements are directly linked to the stimulus intensity evoked by increasing pressures of a balloon inserted into the distal colon. Strikingly, compared to uninfected mice, both *per os* and ip-infected mice exhibited a significant reduction of VMR in response to colorectal distension (Fig 1D). Hence, *T. gondii* chronic infection decreases visceral nociceptive responses regardless of the mode of infection, and thus regardless of local parasite persistence (as parasite load was undetectable in the colon of *per os* chronically infected mice, see Fig 1C). Since visceral hypoalgesia was similarly induced following *per os* and ip routes of infection, and since ip infections avoid the use of 'reservoir' mice required to prepare the cysts, we conducted the rest of the study with ip-infected mice.

To start deciphering the mechanisms explaining this hypoalgesia, we assessed the presence of potential anomalies of the colonic tissue at macroscopic and microscopic levels. Colon thickness and length were not changed at 10 weeks post-infection, showing that this model of chronic latent infection does not induce a macroscopic long-lasting colonic inflammation (Fig 1E). Moreover, H&E stainings of the colon showed that chronically infected mice have a normal architecture and do not display any microscopic damage or massive immune infiltration (Fig 1F). Together, these data indicate that independently from the route of infection, latent *T. gondii* infection decreases visceral nociception in the absence of apparent gut inflammation.

### *Toxoplasma gondii* chronic infection induces long-lasting modifications of the colonic immune landscape

In line with the fact that *T. gondii* infection has not been reported to be a trigger of PI-IBS, mice chronically infected with *T. gondii* exhibited lower visceral response to colorectal distension. This result suggests that analgesic pathways are mobilized in *T. gondii* infected mice, possibly explaining the diminished VMR observed upon chronic infection. Several studies have highlighted the role of immune cells and mediators in modulating visceral sensitivity and pain [36–39]. Therefore, we investigated potential changes in colonic immune infiltration during *T. gondii* infection.

First, we set out to analyze the main immune cell modifications occurring in the colon shortly after infection (i.e., at acute stage). To this aim, we performed an unsupervised flow cytometry analysis of colonic cells extracted from the colon of non-infected mice vs mice infected with *T. gondii* for 14 days. This analysis indicated that the colon of acutely infected mice is enriched in both innate and adaptive immune cells (S1 Fig). More specifically, the colon of acutely infected mice exhibited a higher proportion of monocytes (cluster 3) and neutrophils (cluster 1) compared to non-infected mice (S1A and S1B Fig). In addition, various T cell subsets were enriched (S1C and S1D Fig), such as IFN-γ-producing CD4+ T cells (clusters 4 & 15), cytotoxic CD8+ T cells (clusters 7 and 2) and T cells producing both IFN-γ and Granzyme B (clusters 5, 6 & 8). In contrast, we found a lower percentage of regulatory (FoxP3+) CD4+ T cells (cluster 3) and of IL-17-producing conventional (FoxP3-) CD4+ T cells (cluster 9) in acutely infected compared to non-infected mice. These data confirmed that acute infection by *T. gondii* is associated with a type 1-polarized immune response, and they allowed us to identify the immune cell subsets of the colon which proportions are most affected (negatively or positively) following *T. gondii* infection.

We next conducted supervised flow cytometry analyses to investigate how the abundance of such immune cell subsets changed throughout chronic infection. The gating strategies used to identify the different immune cell subsets are depicted in S2 Fig. First, in contrast to acute infection, we observed no change in the number of leukocytes (i.e., CD45+ Epcam-negative) present in the colon of chronically infected mice compared to non-infected mice, indicating that immune infiltration

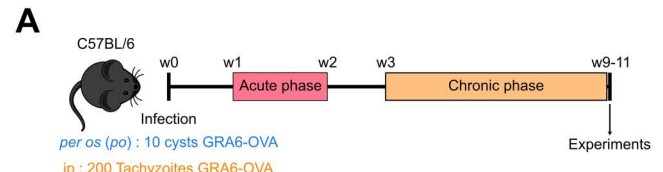

**Fig 1. *Toxoplasma gondii* chronic infection decreases visceral nociceptive responses in the absence of macroscopic colonic inflammation.**
**(A)** Experimental workflow: C57BL/6 mice were infected with Pru.GFP.GRA6-OVA *T. gondii* either *per os* with 10 cysts or by intraperitoneal injection with

200 tachyzoïtes. Experiments were performed between 9 and 11 weeks post-infection. Cartoon was modified from https://openclipart.org/detail/17558/simple-cartoon-mouse. **(B)** Weight measurements during the course of infection, relative to the initial body weight. Dots represent the median +/- IQR of n = 37 (ni), 22 (ip), or 29 (*per os*) mice. **(C)** Parasite burden per µg of tissue DNA measured by qPCR on genomic DNA extracted from brain and colon. Box and whisker plots show median + /- IQR. Data are from 3 pooled independent experiments with each dot representing one mouse. Statistical analysis was performed with Kruskal-Wallis test and subsequent Dunn's correction for multiple comparisons. **(D)** Visceromotor response (VMR) to increasing colorectal distension pressure (15 to 60 mm Hg) measured in non-infected (ni) mice or mice infected either *per os* or ip. Data are shown as mean + /- SEM and correspond to 3 pooled experiments with n = 15 (ni), 6 (*per os*), and 12 (ip) mice per group. Statistical analysis was performed with Areas Under the Curve (AUC) using a Kruskal-Wallis test with Dunn's correction for multiple comparisons. **(E)** Colon length was measured and colon was opened longitudinally to measure thickness. Data are pooled from 2 independent experiments. Statistical analysis was performed using one-way ANOVA. **(F)** Representative pictures of Hematoxylin/Eosin staining of distal colon section (5 µm thick) from 5 ni vs. 8 ip-infected mice (left pictures) or 5 ni vs. 6 per os-infected mice (right pictures). Scale bar = 200 µm.

of the colon has fully resolved at chronic stage (Fig 2A), in agreement with the normal histology described above (see Fig 1E and 1F). This observation was corroborated by the similar percentages of monocytes, neutrophils and Tregs in the colon of non-infected *vs*. chronically infected mice (Fig 2A). Yet, the composition of the colonic immune compartment remained durably changed upon infection since infected mice showed a significant increase in the proportion of macrophages (F4/80+, CD11b+, Ly6C/G-), conventional CD4+ T cells (Foxp3-) and both TCRαβ+ CD8+ and TCRγδ+ CD8+ T cells (Fig 2A). In addition, T cells from infected mice were more prone to produce effector molecules after PMA/ionomycin restimulation, as demonstrated by the higher number of IFN-γ-positive conventional CD4+ T cells and TCRαβ CD8+ T cells and the higher number of granzyme B-positive (reflecting cytotoxicity) TCRαβ+ CD8+ and TCRγδ+ CD8+ T cells in the colon of *T. gondii*-infected mice (Fig 2B and 2D). In conclusion, while the proportions of neutrophils, monocytes and Tregs out of CD45+ leukocytes normalized over time, chronic infection by *T. gondii* induced long lasting modifications of the colonic immune landscape, characterized by an enrichment in macrophages, cytotoxic CD8+ T cells and IFN-γ-producing CD4+ and CD8+ T cells, illustrating the prolonged activation of several T cell subsets beyond the acute infection.

### *Toxoplasma gondii* infection induces an accumulation of colon-resident memory T cells (Trm)

The above data indicate that chronic *T. gondii* infection is associated with a long-lasting increase of colonic T cells with enhanced effector functions. Since the parasite is cleared from the colon in most mice during chronic phase (see Fig 1C), these data cannot be explained by continuous stimulation of T cells with parasite-derived antigenic peptides in the colon. Instead, this increase could result from the generation and persistence of resident memory T cells in the colon throughout chronic phase. In the brain, it is now well established that *T. gondii* infection promotes the generation of parasite-specific brain-resident CD8+ T cells, which sequentially acquire expression of residency markers such as CD69, CD49a and CD103, allowing their retention in the tissue [35,40]. To investigate whether T cells persisting in the colon of *T. gondii* chronically infected mice also exhibit a tissue-resident memory phenotype, we examined the expression of CD49a by CD4+ T cells and of CD49a and CD103 by CD8+ T cells. CD103 was not considered for CD4+ T cells since it is more a Treg marker than a meaningful marker of resident CD4+ Tconv, and CD69 could not be used since it is induced by the *in vitro* PMA/ionomycin stimulation step. We found that chronic infection by *T. gondii* leads to an increase in the frequency of CD49a-positive cells among both conventional (Foxp3-) CD4+ and TCRαβ+ CD8+ T cells. However, no difference was found in the frequency of CD49a/CD103 double-positive cells among TCRαβ+ CD8+ T cells (Fig 3A and 3B). Since we reported above that *T. gondii* chronic infection causes an elevated production of IFN-γ and granzyme B by T cells in the colon, we wondered whether this increased cytokine production could be linked to the increase in colon-resident T cells, as Trm are typically known as fast responders following stimulation. We noticed that infection significantly increased the abundance of resident (CD49a single positive) TCRαβ+ CD8+ T cells able to produce IFN-γ or granzyme B in response to PMA/ionomycin restimulation, while no statistical difference was observed on CD49a-negative or CD49a/CD103 double-positive cells (Fig 3C



**Fig 2. *Toxoplasma gondii* infection induces long-term increase of colonic Th1 CD4+ T cells and cytotoxic CD8+ T cells. (A-D)** Flow cytometry analysis of the indicated cell subsets in the colon of non-infected (ni) vs. mice chronically ip-infected for 70 days (*T. gondii*). Box and whisker show

median +/- IQR. Data are pooled from 2 independent experiments and each dot represents one mouse with a total of 12 (ni) and 11 (*T. gondii*) mice. Statistical analysis was performed using Mann-Whitney test or unpaired Student's t test depending on the normality of the values. **(A)** Box and whisker plots show absolute number of CD45+ cells and percentage of immune cell subsets out of total CD45+ cells in the colon of mice infected (orange) or not (white) for 70 days. **(B-D)** Representative FACS plot for each condition with numbers (in blue) indicating the median percentage +/- IQR of positive cells among **(B)** conventional (Foxp3-) CD4+T cells, **(C)** TCRαβ CD8+T cells and **(D)** TCRγδ CD8+T cells. Box and whisker plots show the absolute number (median +/- IQR) of each T cell subset in the colon of non-infected (white) vs infected (orange) mice.

and 3D). Similarly, *T. gondii* latent infection caused an increase in conventional CD4+ T cells producing IFN-γ upon PMA/ionomycin restimulation, with the most prominent increase observed among the resident CD49a-positive subset (Fig 3E and 3F). Combined, these data indicate that chronic infection by *T. gondii* generates CD49a+ Trm that persist in the colon and display heightened effector functions.

Since inflammation is often associated with pain, this increase in 'pro-inflammatory' type 1-polarized resident T cells could appear antagonistic with the observed decreased visceral nociceptive responses of chronically infected mice. Importantly, studies have highlighted that effector T cells, in particular CD4+ T cells, can release endogenous opioid peptides called enkephalins, that are able to alleviate pain in different inflammatory contexts, including colitis. Furthermore, it was reported that enkephalins could be produced by colitogenic T cells, highlighting the dual effect of T cells on inflammation and pain modulation [41,42]. Notably, transcripts of *Penk*, the gene coding for enkephalins, were the only detectable mRNA of endogenous opioids in the colon tissue (S3A Fig). Based on this, we hypothesized that the release of enkephalins by T cells present in the colon of chronically infected mice may be responsible for the decrease in visceral nociceptive responses.

### *Toxoplasma gondii* chronic infection decreases visceral nociceptive responses through peripheral opioid-signaling

To assess the contribution of enkephalins produced by T cells in the hypoalgesic phenotype caused by latent *T. gondii* infection, we took advantage of a new mouse model enabling constitutive T cell-specific deletion of the pro-enkephalin-encoding gene (Penk). To this aim, we crossed Penk-flox mice [43] with CD4-Cre mice. Since most T cells are derived from a thymic precursor stage that co-expressed CD4 and CD8, this CD4-Cre model should allow the deletion of Penk in both mature CD4+ and CD8+ T cells (S3A Fig). We confirmed by FlowFISH that compared to CD4-Cre- mice, CD4-Cre+ mice display a 10-fold decrease in the percentage of T cells expressing Penk mRNA (S3B Fig). Using this model in which T cells are able (Penk T$^{WT}$) or not (Penk T$^{KO}$) to produce enkephalins, we assessed visceral nociceptive responses to colorectal distension in chronically infected mice (S3C Fig). Preventing enkephalin production by T cells did not change the visceromotor response of chronically infected mice, suggesting that T cell-released enkephalin peptides are not involved in the regulation of visceral nociception in the context of chronic infection by *T. gondii* (S3D Fig). This negative result is corroborated by FlowFISH data showing that *Penk* gene expression is not increased (it is in fact rather decreased) in T cells from the colon during latent *T. gondii* infection (S5 Fig). Notably, these results suggest the potential contribution of other cellular sources of enkephalins, or even other types of opioids, in the hypoalgesic phenotype. To formally assess the implication of opioid signaling, we tested whether the hypoalgesic phenotype could be reversed by blocking opioid receptors. To restrict the inhibition to peripheral opioid receptor, we used naloxone methiodide, an opioid receptor antagonist which cannot cross the blood brain barrier [44]. We assessed visceral nociception in chronically infected mice before and after naloxone methiodide injection (Fig 4A). As shown in Fig 4B and 4C, the administration of naloxone methiodide to infected mice following both per os and ip infection, significantly increased nociceptive responses, indicating that peripheral opioid receptor signaling is involved in the hypoalgesic phenotype induced by *T. gondii* chronic infection, independently from the infection route. In line with previous findings [37,41,45], the same naloxone methiodide

**A**

Gated on conventional CD4+ T cells



**B**

Gated on TCRαβ CD8+ T cells

**C**

Gated on TCRαβ CD8+ T cells subsets

**D**

**E**

Gated on conventional CD4+ T cells subsets

**F**

**Fig 3. *Toxoplasma gondii* infection generates Th1 CD4+ and cytotoxic CD8+ resident T cells. (A-F)** FACS plots or Box and whisker plots with median +/- IQR showing absolute number or percentage of cell subsets in the colon of non-infected (ni) or chronically ip-infected mice (*T. gondii*) for



70 days. Data are pooled from 2 independent experiments and each dot represents one mouse with n = 12 (ni) *vs.* 11 (*T. gondii*). Depending on the normality of the dataset, statistical analysis was performed using Mann-Whitney test or unpaired t test when comparing 2 conditions **(A and B)**, or using Kruskal-Wallis test with Dunn's correction for multiple comparisons **(D and F)**. **(A and B)** Representative FACS plots for each condition with numbers in blue showing median percentage +/- IQR of CD49a-positive cells among **(A)** conventional (Foxp3-) CD4+ T cells and **(B)** TCRαβ+ CD8+ T cells. Corresponding frequencies of each T cell subset are represented with box and whisker plots showing median +/- IQR. **(C and D)** Representative FACS plot for each condition with numbers in blue indicating the median percentage +/- IQR of cytokine-positive cells out of TCRαβ CD8+ T cells **(C)** and the absolute number of each subset **(D),** represented as box and whisker plots showing median +/- IQR. **(E and F)** Representative FACS plots for each condition with numbers in blue indicating the median percentage +/- IQR of cytokine-positive cells out of conventional (Foxp3-) CD4+ T cells **(E)** and the absolute number of each subset **(F),** represented as box and whisker plots showing median +/- IQR.

treatment did not change the visceral nociception of uninfected mice (S4 Fig). Therefore, collectively, our data show that peripheral opioid receptor signaling underlies the decreased visceral nociceptive responses observed upon chronic infection by *T. gondii* but that enkephalins produced by T cells do not play a major role in this phenotype.

### Latent chronic infection by *T. gondii* induces moderate changes in the gut microbiota and in the inferred gut microbiome

To understand whether the phenotypes related to both colon immune modulation and decreased visceral nociceptive responses induced by *T. gondii* may be associated with a change in colon microbiota, we analyzed both the microbial profiles and inferred microbial functions in the colon mucosa of non-infected and *T. gondii*-infected mice. The general microbial taxonomic profile of both groups of mice was not significantly different, most likely due to a high intragroup variance that was mainly observed in the non-infected group (S6A Fig). More specifically however, 2 bacterial species, i.e., *Odoribacter splanchnicus* and *Lactobacillus reuteri,* displayed a lower relative abundance in infected compared to non-infected control mice (S6B Fig). Accordingly, chronic infection by *T. gondii* induced a general significant decrease in most of the diversity indices analyzed, mostly driven by a significant decrease in ACE (related to community richness) and both iChao and Chao indices (related to community diversity driven by rare species) (S6C Fig). In terms of inferred microbial functions, the general microbial functional profiles at 3 levels, i.e., pathways, Kyoto Encyclopedia of Genes and Genomes (KEGG) and Enzyme Commission (EC), of both groups of mice were not significantly different, again most likely due to a high intragroup variance (S6D, S6F and S6G Fig). However, 2 pathways, i.e., rhamnose catabolism and nitrate assimilation showed a lower relative abundance in infected compared to non-infected control mice (S6E Fig). Overall, these data suggest that latent chronic infection by *T. gondii* induces a moderate taxonomical and functional decrease in the microbiota associated with the colon mucosa.

## Discussion

*T. gondii* is a widespread parasite that establishes chronic infection in all warm-blooded animals including humans. While *T. gondii* infection is known to have major consequences on brain functions, the long-term impact of this infection on the gut compartment remains ill-defined. Using a mouse model of latent *T. gondii* infection, this study reports that latent infection downregulates basal visceral nociceptive response measured to colorectal distension. This phenotype was observed both after natural infection (*per os*) and following intraperitoneal administration of tachyzoites. Experimental models suggest that the primary site of host invasion, before disseminating to other tissues, is the small intestine [46–48], but not the colon. Therefore, we hypothesize that the hypoalgesic phenotype detectable at chronic stage in the colon is more likely to result from secondary events following systemic dissemination than to originate from primary lesions/changes related to initial host invasion. In that sense, the tachyzoite ip model appears adequate to reproduce this setting. Importantly, we observed that this decreased visceral nociceptive response is dependent on peripheral opioid receptor signaling. Whether the parasite is directly responsible for the desensitization of nociceptive fibers was not formally addressed in our work. However, we think it is unlikely since the hypoalgesic phenotype was observed in a context where *T. gondii* has

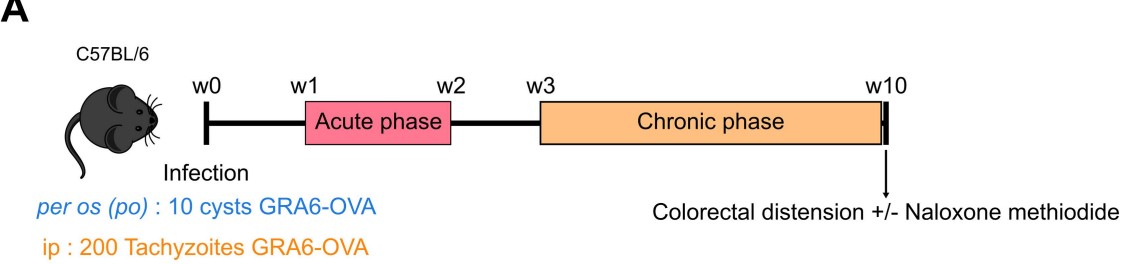

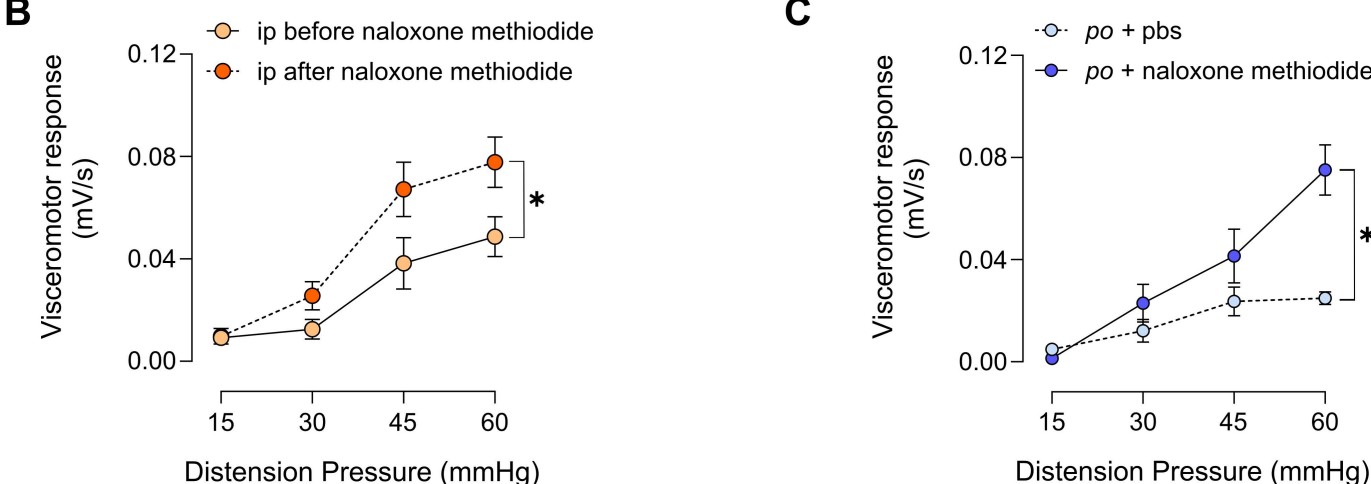

**Fig 4. Decreased visceral nociceptive responses induced by *Toxoplasma gondii* chronic infection relies on peripheral opioid signaling. (A)** Experimental workflow: C57BL/6 mice were infected with Pru.GFP.GRA6-OVA *T. gondii* either *per os* with 10 cysts or by intraperitoneal injection with 200 tachyzoïtes. At 10 weeks post-infection, colorectal distension was performed to assess visceral nociception of each mice 30 min after intraperitoneal injection of naloxone-methiodide or PBS. Cartoon was modified from https://openclipart.org/detail/17558/simple-cartoon-mouse. **(B)** Visceromotor response (VMR) to increasing colorectal distension pressure (15 to 60 mmHg) was measured in ip-infected mice before (in orange) or after (in red) naloxone methiodide injection. Data are shown as mean +/- SEM and are from 1 experiment with n = 10 mice per group. Statistical analysis was performed using a paired t test with total area under the curve. **(C)** Visceromotor response (VMR) to increasing colorectal distension pressure (15 to 60 mmHg) was measured in *per os* infected mice injected with naloxone methiodide or PBS 30 min before the colorectal distension. Data are shown as mean +/- SEM and are from 1 experiment with n = 6 mice per group. Statistical analysis was performed using a Mann-Whitney's test with total area under the curve.

been cleared from peripheral tissues and the only reservoir of *T. gondii* is the CNS. By analogy with the current view that neuroinflammatory pathways, more than *T. gondii* cysts themselves, underlie the *T. gondii*–mediated cognitive alterations [26,49,50], we suspect that the consequences of chronic *T. gondii* infection on colon pain perception result from local neuroimmune imprinting, ultimately leading to a durable shift of the balance towards anti-nociceptive signals, rather than from a direct manipulation of infected neurons by *T. gondii* in the colon or DRG.

In our view, this hypoalgesic phenotype was unexpected for at least two reasons. First, *T. gondii* infection has so far been reported to be an aggravating factor rather than a negative modulator of intestinal inflammation and immunopathology, two conditions that are classically associated with pain. Along this line, one study in humans has revealed a higher seroprevalence of *T. gondii* in Crohn's disease patients, although not in Ulcerative Colitis patients [51]. In experimental models, analyses of orally infected C57BL/6 mice showed massive inflammatory lesions of the ileum during acute phase [52,53], suggesting that *T. gondii* could actually be an infectious trigger of inflammatory bowel disease (IBD) [54,55].

Accordingly, chronic *T. gondii* encephalitis was recently found to exacerbate the susceptibility of mice to DSS-induced colitis, an established model of IBD [28]. Data from our paper actually support this hypothesis since we observed that latent *T. gondii* infection mobilizes pro-inflammatory T cells in the colon, including resident T cells, which are known to contribute to chronic intestinal inflammation [56].

Yet, IBD and IBS are notably distinct pathologies. While IBD is characterized by severe and chronic inflammation of the GI tract, IBS is mostly a functional syndrome that is associated with low-grade inflammation or immune activation of the gut, and is generally devoid of overt chronic inflammation. Nevertheless, several gastrointestinal infections have been reported to cause PI-IBS [17,57], which is the second reason why our data are unexpected. To our knowledge, only one case-control study explored the association between *T. gondii* exposure and visceral sensitivity [58]. In this study, the seroprevalence of *T. gondii* infection was higher in subjects with frequent abdominal pain (8%) than in control subjects (4%), suggesting an association between *T. gondii* infection and frequent abdominal pain. However, the context of abdominal pain (e.g., IBS, pharmacological treatment and/or infection) was not described, thereby hampering interpretation of this work. Moreover, since it is unknown whether seroconversion preceded or not the onset of abdominal pain, our understanding of the underlying mechanisms remains limited. A serodiagnosis of *T. gondii* exposure in well-characterized cohorts of IBS patients, with a detailed description of clinical symptoms and visceral sensitivity, would be needed to establish to which extent our data translate to the human setting.

An important mechanistic achievement of our study is to show that opioid receptor signaling in peripheral neurons underlies *T. gondii*-induced hypoalgesia of the colon but that T cell-produced enkephalins play only a limited or redundant role in this phenotype. Furthermore, the level of *Penk* gene expression is globally reduced in various colon-isolated cells during latent infection as assessed by FlowFISH. Several hypotheses can be envisaged to reconcile the apparent paradox between an opioid receptor-mediated hypoalgesia and a reduction in Penk mRNA in colon-isolated cells: (i) various cell types are able to produce opioids, especially enteric neurons which may not be as amenable for isolation and flow cytometry analysis as lymphocytes. A heightened opioid secretion by such cells, or by other cellular subpopulations not evaluated here, may underlie the increased opioid receptor activity; (ii) independently from mRNA expression level, hypoalgesia may ultimately arise from post-translational changes of opioid peptide precursors leading to increased processing and/or increased secretion of the final bioactive peptides. Quantifying changes in bioactive opioid peptides in the colon microenvironment upon chronic *T. gondii* infection would be useful but the absence of reliable tools to perform this at the tissue and single-cell levels was a major roadblock preventing us from tackling this question; (iii) at last, a closer proximity between opioid-producing cells and nociceptive fibers of the colon may also contribute to reinforce the biological effects of these labile molecules.

Visceral nociception results from the integration of both pro- and anti-nociceptive signals in the gut, and thus can be modulated by changes in activity of the opioid receptor signaling pathways expressed by sensory neurons. Therefore, an alternative possibility is that other (non-opioid) molecules may be involved, provided that such molecules are capable of modulating the opioid receptor signaling pathway. Of note, serotonin (5-hydroxytryptamine, 5-HT) is a pro-nociceptive mediator able to directly activate neurons [59]. In the GI tract, 5-HT is mainly produced by enterochromaffin cells (EC cells), a subset of enteroendocrine cells located within the epithelium. Importantly, EC cells transduce luminal signals to mucosal nerve endings, producing various effects including nociceptor sensitization leading to visceral hypersensitivity [60]. Moreover, several studies have reported alteration of serotonin metabolism in IBS patients [61,62], suggesting that serotonin signaling could alter pain perception in the gut. In line with these observations, a recent study highlighted a link between serotonin and opioid signaling since both pathways act on PKA activity, though in an opposite manner. More precisely, a decreased serotonin signaling was associated with an enhanced opioid receptor signaling, resulting in a prolonged endogenous analgesia [63]. Serotonin release by EC cells can be triggered by various stimuli, including short chain fatty acids produced by the gut microbiota or neurotransmitters [59], which may be dysregulated upon *T. gondii* chronic infection. Elucidating the potential implication of the serotoninergic pathway in *T. gondii*-induced hypoalgesia could be an interesting avenue for future investigation.

Based on the abundance of the microbiota in the colon and the pleiotropic effects exerted by gut microbiota on the host including the production of 5-HT [64] and the modulation of opioid receptor-mediated analgesia [65], another attractive hypothesis is that changes in visceral nociception caused by chronic *T. gondii* infection may arise from long-lasting alteration of gut commensals. To date, experimental studies have shown that acute *T. gondii* infection causes a clear dysbiosis in the colon, characterized in particular by an increase in *Enterobacteriacea* [29,66]. However, the extent of colon dysbiosis during chronic infection appears more variable, with studies reporting either no impact on gut microbiota [28,29] and studies showing mild modifications in the *Bacteroidetes* and *Firmicute* phyla [30,31]. In our study, we found a minor taxonomical and functional decrease in the colon mucosa-associated microbiota of chronically infected mice, compared to uninfected mice. While *T. gondii* chronic infection does not induce dramatic changes in the colon mucosa microbiota, we cannot rule out the possibility that subtle alterations in the abundance of some species might be sufficient to modulate visceral nociceptive responses in our mouse model. Notably, a study has highlighted the capacity of a particular bacterial species, *L. reuteri*, to upregulate opioid receptor expression, thereby preventing visceral hypersensitivity in a rat model of colon obstruction [65]. In contrast, we found that the hypoalgesic phenotype is associated with a decrease in *L. reuteri* relative abundance. This apparent discrepancy may be explained if the contribution of *L. reuteri* to visceral sensitivity is dependent on both i) its anatomical site, i.e., colon feces (the cited study) vs colon mucosa (our study) and ii) the pain model.

In conclusion, our work not only provides the first detailed mapping of immune changes that persist in the colon throughout latent infection, but it also uncovers a new and unexpected consequence of *T. gondii* chronic infection on host intestinal homeostasis. We report that *T. gondii* latent infection elicits a prolonged modification of the colonic T cell compartment as well as a decreased visceral nociceptive response to colorectal distension. This hypoalgesia depends on peripheral opioid receptor signaling but is independent from enkephalins produced by T cells, suggesting that other cell types and/or other opioids may be involved. Our study suggests that *T. gondii* is unlikely to be a major contributor of post-infectious IBS and may instead indicate a potential negative association between latent *T. gondii* infection and abdominal pain.

## Materials & methods

### Ethics statement

Animal care and used protocols were carried out under the control of the French National Veterinary Services and in accordance with the current European regulations (including EU Council Directive, 2010/63/EU, September 2010). The protocols APAFIS 20921–2019052311562282 and APAFIS 14513-2018040313435341v6 were approved by the local Ethical Committee for Animal Experimentation registered by the "Comité National de Réflexion Ethique sur l'Experimentation Animale" under no. CEEA122.

### Mice

C57BL/6 (B6) and CBA/J mice were purchased from Janvier (France). PenkT$^{KO}$ mice were obtained by crossing Penk$^{flox/flox}$ mice [43] with CD4-Cre mice (B6.Cg-Tg(Cd4-cre)1Cwi/BfluJ from JAX, ref 022071). All non-commercial mouse models were housed and bred under specific pathogen-free conditions at the 'Centre Regional d'Exploration Fonctionnelle et de Ressources Expérimentales' (CREFRE-Inserm UMS006). Mice were housed under a 12h light/dark cycle and were experimentally infected between 8 and 10 weeks of age. All mice used in experiments were males and mice with undetectable parasite load in the brain (measured by qPCR on genomic DNA after euthanasia) were excluded from the study. Number of mice and experimental replicates are indicated in the respective figure legends.

### *Toxoplasma gondii* culture and experimental infections

Human Foreskin Fibroblasts (HFF) were cultured in Dulbecco's Modified Eagle Medium (DMEM, Gibco) supplemented with Glutamax, sodium pyruvate, 10% vol/vol FBS (Gibco), 1% vol/vol Penicillin/Streptomycin (Gibco) and 0.1% vol/vol

2-β-mercaptoethanol (Gibco). Pru-GFP.GRA6-OVA tachyzoites were grown *in vitro* on confluent HFF in DMEM supplemented with 1% vol/vol FBS (Gibco). *In vitro* cultured parasites were filtered through a 3 μm polycarbonate hydrophilic filter (it4ip S.A.) and diluted in sterile PBS before infection of mice with 200 tachyzoïtes. For cyst generation, CBA/J mice were infected by intraperitoneal injection with 200 tachyzoïtes. After 2 months, brains of infected CBA mice were collected and homogenized before performing a rhodamine-conjugated Dolichos Biflorus Agglutinin (Vector Laboratories) labelling. Cysts were counted using an inverted fluorescence microscope. After cyst enumeration, brain homogenate was diluted in sterile PBS to orally infect mice by gavage with 10 cysts.

### Parasite load quantification (DNA extraction, qPCR)

Genomic DNA (gDNA) was extracted from tissue homogenate using the DNeasy Blood & Tissue Kit (Qiagen). All tissues analyzed were snap frozen in liquid nitrogen directly after isolation before storage at −80 °C. For brain and colon, tissues were kept frozen in liquid nitrogen and crushed to obtain a fine powder used to isolate gDNA according to manufacturer's recommendations. gDNA was quantified using a Nanodrop and the number of parasite per μg of tissue DNA was calculated by qPCR using the TOX9/TOX11 primers and a standard curve made of a known number of corresponding parasites (as described in [67]). The quantification threshold indicates the highest dilution of the standard curve, above which parasite concentration could be extrapolated.

### Quantification of gene expression by RT-qPCR

Colon homogenate was obtained using a Precellys and 1.4 mm ceramic beads (MpBio). Total RNA was then extracted from the tissue lysate using a phenol chloroform-based extraction and the Direct-Zol RNA kit. RNA was quantified using Nanodrop and reverse transcription was performed using the SensiFAST cDNA synthesis kit (Bioline Meridian) according to the manufacturer's recommendations. qPCR was performed using a LightCycler 480 System (Roche) following an initial denaturation at 95 °C for 5 min followed by 45 cycles of (15 s at 95 °C–20 s at 60 °C–20 s at 72 °C). Cycle threshold (Ct) was extracted from the graphs using the LightCycler 480 software. The following forward and reverse primers were used: 5'-CTGGTTAAGCAGTACAGCCCCAA-3' and 5'- CGAGAGGTCCTTTTCACCAGC-3' for *H*prt; 5' CGACATCAATTTCCTG-GCGT 3' and 5' AGATCCTTGCAGGTCTCCCA 3' for *Penk;* 5' TGGCCCTCCTGCTTCAGAC 3' and 5' CAGCGAGAGGTC-GAGTTTGC 3' for *Pomc*; 5' TGTGTGCAGTGAGGATTCAGG 3' and 5' AGACCGTCAGGGTGAGAAAAGA 3' for *Pdyn*. Gene expression was quantified by calculating the difference of Ct between the housekeeping gene (*Hprt*) and the gene of interest (*Penk, Pomc* or *Pdyn*).

### Hematoxylin/Eosin (H/E) stainings

Colonic tissues were fixed for 24h in 4% formaldehyde and then transferred in 70% ethanol at 4 °C. Tissues were embedded in paraffin to perform tissue sections (5 μm thickness) and Hematoxylin/Eosin stainings.

### Isolation of colonic cells & PMA/ionomycin restimulation

After mouse euthanasia, the colon was collected, opened longitudinally and kept on ice in RPMI supplemented with 10% FBS. Tissue was cut into small pieces before incubation in RPMI supplemented with 5% FBS and 500mM EDTA to remove epithelial cells and intraepithelial lymphocytes (IEL) for 30 min at 4 °C under agitation. Supernatant (containing IEL) was filtered through 40 μm cell strainer, washed once and kept on ice until Percoll gradient. The remaining tissue was then digested in RPMI supplemented with 10% FBS, 1 M HEPES, 220 U/mL Collagenase VIII (Sigma) and 10 μg/mL DNase I (Sigma) for 1 hour at 37 °C under agitation. The supernatant was collected, filtered through a 70 μm cell strainer, washed and pooled with the IEL fraction, and kept on ice. The remaining tissue was digested a second time with the same digestion buffer for 50 min at 37 °C under agitation. At the end, tissue was mechanically dissociated with a 18G needle

attached to a 5 mL syringe. Cells were filtered through a 70 µm cell strainer before being pooled with the other fractions. Cells were then isolated using a 30% Percoll gradient diluted in RPMI at 4 °C. Cell suspension was used directly for flow cytometry staining, or restimulated for 3.5 h at 37 °C 5% $CO_2$ in complete RPMI supplemented 0.05 µg/mL Phorbol 12-myristate 13-acetate (PMA), 1 µg/mL ionomycin and 3 µg/mL Brefeldin A (BfA). Cells were then washed with FACS buffer (2% FBS and 2mM EDTA) before performing flow cytometry stainings.

### Flow cytometry stainings & analysis

For flow cytometry, after Fc receptor saturation and dead cell detection with fixable viability dye, cells were surface stained for 30 min at 4 °C. Cells were then washed and fixed using the Foxp3/transcription factor staining buffer set (Invitrogen) before performing intracellular staining to detect cytokines and transcription factors according to the manufacturer's recommendations. Flow cytometry analyses were performed using FlowJo and OMIQ (unsupervised analyses) softwares.

### Antibodies used in this study

CD8α-BV786 & CD8α-BV421 (clone 53-6.7), CD11b-PE-CF594 (clone M1/70), IFN-γ-BV421 (clone XMG1.2), NK1.1-BV650 (clone PK136), RORγt-BV421 (clone Q31-378), CD44-BV605 (clone IM7), CD4-PE-CF594 (clone RM4–5 from BD Horizon, CD49a-BUV737 (clone Ha31/8) from BD OptiBuild; CD4-APC-Cy7 (clone GK1.5), CD11c-PerCP-Cy5.5 (clone HL3), IL-17-AF700 (clone TC11-18H10), CD62L-PE-Cy7 (clone MEL-14) from BD Pharmingen; anti-mouse CD16/62 (clone 93), CD3ε-BV711 & CD3ε-APC (clone 145-2C11), CD19-APC (clone 6D5), CD45-PE (clone 30-f11), CD45.2-AF488 (clone 104), CD117 (c-kit)-BV785 (clone 2B8), EpCAM-APC & EpCAM-PE-Dazzle594 (clone G8.8), F4/80-AF700 (clone BM8), FcεRI-APC-Cy7 (clone MAR-1), granzyme B-PE-Cy7 (clone QA16A02), Ly6C-BV711 (clone HK1.4), Ly6G-BV510 (clone 1A8), TCRγδ-BV510 (clone GL3) from Biolegend; Fixable Viability Dye-eFluor660, CD90.2-PE-Cy7 (clone 53-2.1), CD103-PerCP-eFluor710 (clone 2E7), Foxp3-PE (clone NRRF-30), Gata3-AF488 (clone TWAJ) from eBioscience.

### FlowFISH stainings

Spleen from Penk$^{flox/flox}$:CD4$^{Cre}$ mice was smashed on a 100 µm cell strainer (Falcon) before performing red blood cell lysis. Splenocytes were then stained with the PrimeFlow RNA Assay Kit (Invitrogen) and the anti-mouse Penk probe set (Invitrogen #PF-204) according to the manufacturer's protocol. Briefly, cells were first labelled with antibodies as described above with the reagent provided in the kit. A second fixation step was performed after the intracellular staining before proceeding to the FISH staining. For FISH staining, cells were incubated with the anti-mouse Penk probe set (1:20) for 2h at 40 °C. After washes, cells were kept overnight at 4 °C in wash buffer containing RNase inhibitors. The day after, amplification steps were performed to increase the signal: cells were first incubated with pre-amplification mix during 1.5 h at 40 °C, washed and then incubated with amplification mix for an additional 1.5 h at 40 °C. Cells were then incubated with the label probes (Alexa Fluor 647, 1:100) for 1h at 40 °C.

### Colorectal distension and visceromotor response measurements

Visceral nociception was assessed by measuring the visceromotor response (VMR) to increasing colorectal distension pressures (15, 30, 45, 60 mm Hg) with at least 5 min rest between each distension pressure. Three days before distension, mice were anaesthetized with ketamine (10 mg/kg) and xylazine (1 mg/kg) to implant 2 electrodes (Bioflex insulated wire AS631; Cooner Wire, Chatsworth, CA) into abdominal external oblique muscles. Electrodes were exteriorized at the back of the neck and protected by a plastic tube attached to the skin. After surgery, mice were monitored during the next 3 days for abnormal behavior. The day of the distension, electrodes were connected to a Bio Amp, itself connected to an electromyogram acquisition system (AD Instruments, Inc., Colorado Springs, CO). Ten (10) second-long distensions were

performed on conscious animals with a 10.5 mm diameter balloon catheter (Fogarty catheter for arterial embolectomy) gently inserted into the distal colon at 5 mm from the anus (balloon covering a distance of 1 cm). Electromyography activity of abdominal muscles was recorded to calculate visceromotor responses using LabChart 8 software (AD Instruments).

### Taxonomic and predictive functional analysis of the colon mucosa-associated microbiota

Colon from uninfected vs latently infected mice was opened longitudinally and placed in ice-cold sterile PBS. Mucosa-associated microbiota was obtained after shaking the colon in sterile PBS. Suspension was then centrifuged 1 min at 100 x g and filtered through a 30 µm cell strainer to remove cell debris. The supernatant, containing the mucosa-associated microbiota, was collected and centrifuged 15 min at 5000 x g at 4 °C. Supernatant was discarded and total DNA was extracted from the pellet as already reported [68]. The 16S rRNA gene V3-V4 regions were targeted by the 357wf-785R primers and analyzed by MiSeq at RTLGenomics (Texas, USA). A complete description of Data Analysis Methodology for Microbial Diversity can be found at https://static1.squarespace.com/static/5807c0ce579fb39e1dd6addd/t/63fd239532130b3602fb9e9b/1677534102629/RTL_Data_Analysis_Methodology_v4.pdf.

In brief, an average of 36551 sequences was generated per sample. Linear discriminant analysis (LDA) score graphs were drawn and predictive functional analysis of the colon mucosa microbiome was performed using the Huttenhower Galaxy web application via the LefSe [69] and PICRUSt2 [70] algorithms, respectively. Principal component analysis (PCA) was performed and diversity indices were calculated using PAST4.17c (Paleontological statistics software package for education and data analysis. https://palaeo-electronica.org/2001_1/past/past.pdf).

### Statistical analysis

Statistical analysis was performed using GraphPad Prism 10.1 software. For all panels, normality was assessed with D'Agostino & Pearson tests. When normal, a parametric test was applied as indicated in the figure legend. If not, a non-parametric analysis was performed. * or #: $p < 0.05$, **: $p < 0.01$, ***: $p < 0.001$, ****: $p < 0.0001$. Modalities of data representation and statistical tests are indicated in the respective figure legends. Statistical differences in PCA were evaluated with 1-way perMANOVA non-parametric test.

## Supporting information

**S1 Fig. Acute *T. gondii* infection induces recruitment of myeloid cells and T cells in the colon.** (**A and B**) Unsupervised analysis with flow cytometry data obtained from the colon of non-infected or ip-infected mice at acute stage (14 days post-infection) was performed using OMIQ software. Statistical analysis was performed using a Mann-Whitney test on each subpopulation and summarized into one single graph. (**A**) Unsupervised analysis was performed on live CD45+ cells negative for CD3ε, CD19 and EpCAM (called Lin-negative cells). Opt-SNE plot of 34 000 Lin-negative cells concatenated from 4 non-infected mice (left) and 42 500 Lin-negative cells concatenated from 5 infected mice (right), divided in 9 clusters using Phenograph algorithm, based on the expression of surface markers associated with each cell subset. Table under opt-sne shows the relative abundance of each cluster in each condition (infected (orange) or not (white)). (**B**) Unsupervised analysis was performed on live T cells using the following gating strategy: EpCAM-/ CD45+/ CD3+. Opt-SNE plot of 36 360 colonic T cells concatenated from 4 non-infected mice (left) and 45 450 colonic T cells concatenated from 5 infected mice (right), divided in 15 clusters using Phenograph algorithm, based on the expression of different markers. Table under opt-sne shows the relative abundance of each cluster in each condition (infected (orange) or not (white)). (PDF)

**S2 Fig. Gating strategies used to identify the immune populations in the colon. (related to Fig 2)** (**A and B**) Cell doublets were excluded using double gating on SSC-A vs. SSC-H followed by SSC-W vs. SSC-H exclusion. Dead cells

were removed using a viability marker to allow analysis of live cells only. Arrows indicate the order of the gating strategy. (**A**) Gating strategy used to identify the different T cell subsets in the colon. (**B**) Gating strategy used to identify the different cell subsets belonging to the myeloid and Innate Lymphoid Cells (ILC) lineages.
(PDF)

**S3 Fig.** *T. gondii*-**induced decrease in nociceptive responses is independent of T-cell derived enkephalins. (related to Fig 4) (A)** Expression of the 3 opioid encoding - genes (i.e., *Pomc, Penk* and *Pdyn*) was measured by RT-qPCR in the whole colon of mice chronically ip-infected for 70 days. Values for gene expression were calculated based on the difference between the housekeeping gene *Hprt* and the gene of interest (ΔCt). Data correspond to a pool of 2 independent experiments and each dot represents one mouse. (**B**) Mouse model for T cell-specific deletion of enkephalin-encoding gene (*Penk*) in mice. *Penk*-floxed mice carrying LoxP sites upstream and downstream of *Penk* exon 2 were crossed with transgenic mice expressing the Cre recombinase under the control of CD4 promoter (CD4-Cre). Cartoon was modified from https://openclipart.org/detail/17558/simple-cartoon-mouse. (**C**) Deletion of Penk was confirmed by measuring the expression of Penk mRNA by flow cytometry (FlowFISH) in naïve (CD62L + /CD44-) or activated (CD44+) conventional (FoxP3-) CD4 + T cells, and in regulatory CD4 + T cells (Foxp3+). (**D**) Experimental workflow: C57BL/6 expressing (Penk $T^{WT}$) or not (Penk $T^{KO}$) in T cells were infected with the Pru.GFP.GRA6-OVA *T. gondii* by intraperitoneal injection with 200 tachyzoïtes. At 10 weeks post-infection, colorectal distension was performed to assess visceral sensitivity. Cartoon was modified from https://openclipart.org/detail/17558/simple-cartoon-mouse. (**E**) Visceromotor response (VMR) to increasing colorectal distension pressure (15–60 mm Hg) was measured in chronically ip-infected mice invalidated (Penk $T^{KO}$) or not (Penk $T^{WT}$) for Penk gene in T cells. VMR are represented with mean + /- SEM with n = 7 Penk $T^{WT}$ vs 6 Penk $T^{KO}$ animals. Statistical analysis was performed on Areas Under the Curve (AUC) with a Mann-Whitney test using GraphPad Prism. Data correspond to one experiment. (**F**) Parasite loads in the brain were measured by qPCR on genomic DNA for each mouse, showing no difference between Penk $T^{WT}$ and Penk $T^{KO}$ infected mice.
(PDF)

**S4 Fig. Naloxone methiodide treatment has no impact on visceral nociceptive responses at steady state. (related to Fig 4) (A and B)** Colorectal distension was performed to assess visceral sensitivity of each mouse before (in white) or 30 min after (in gray) intraperitoneal injection of naloxone-methiodide. (**A**) Visceromotor response (VMR) in response to increasing colorectal distension pressure (15–60mmHg) was measured in uninfected mice (ni) treated or not with Naloxone methiodide. Data are shown as mean + /- SEM and are from 1 experiment representative with n = 6 mice. (**B**) Area Under the Curve (AUC) are represented for each mouse with 2 dots corresponding to the AUC before (in white) and after (in gray) naloxone methiodide treatment. Statistical analysis was performed using a Wilcoxon's test.
(PDF)

**S5 Fig. Chronic infection by** *T. gondii* **does not increase** *Penk* **mRNA expression in colonic cell subsets.** (**A** and **B**) Combination of flow cytometry analysis and FISH detection of enkephalin encoding mRNA (*Penk* mRNA) in different colonic cell types in non-infected (white) and chronically infected mice (ip, orange). (**A**) Gating strategy used to identify the different cell subsets in the colon. Cell doublets were excluded using double gating on SSC-A vs. SSC-H followed by SSC-W vs. SSC-H exclusion. Dead cells were removed using a viability marker to allow analysis of live cells only. Arrows indicate the order of the gating strategy. (**B**) Flow cytometry analysis of the indicated cell subsets in the colon of non-infected (ni) vs. mice chronically ip-infected for 70 days (*T. gondii*). Box and whisker show median + /- IQR of the percentage of *Penk*-expressing cells in the cell subset indicated above each graph. Data are from 1 experiment and each dot represents one mouse with a total of 5 non-infected (ni) and 7 chronically ip-infected (*T. gondii*) mice.
(PDF)

**S6 Fig. Changes in the colon mucosa-associated microbiota and inferred microbiome functions, induced by *T. gondii* chronic infection. (A-C)** Taxonomic analysis of the colon mucosa microbiota of latently infected (*T. gondii,* 70 dpi) vs. uninfected (ni) mice with **(A)** Euclidean distance-based Principal Component Analysis (PCA); **(B)** linear discriminant analysis (LDA) scores showing the 2 microbial taxa that are significantly enriched in uninfected mice; **(C)** diversity indices. **(D-G)** Inferred functional analysis of the colon mucosa microbiota of latently infected (*T. gondii,* 70 dpi) vs. uninfected (ni) mice with **(D)** Euclidean distance-based PCA; **(E)** LDA scores showing the 2 microbial pathways significantly enriched in uninfected mice; **(F)** Kyoto Encyclopaedia of Genes and Genomes (KEGG)-based PCA and **(G)** Enzyme Commission (EC)-based PCA. In **(C),** statistical analysis was done with 2-way-ANOVA followed by the 2-step linear procedure of Benjamini, Krieger and Yekutieli to correct for multiple comparisons by checking false discovery rate (<0.05). *P<0.05, **P<0.01. In **(A, D, F, G)**, statistical differences in PCA were evaluated with 1-way perMANOVA non-parametric test. Data are from 1 experiment with n=5 mice per group.
(PDF)

## Acknowledgments

We thank R. Balouzat, R. Ecalard, F. Chaboud, E. Debon, M.A. El Manfaloti, S. Negroni, J. Leblond, S. Fresse, M. Lulka from ANEXPLO-CREFRE UMS006 for ethical care of our models, F. L'Faqihi-Olive, V. Duplan-Eche, A.-L. Iscache, H. Garnier from the flow cytometry core facility of Infinity for expert technical assistance, R. Miranda-Capet, E. Valdevit, A. Edir and A. Herrmann for technical help. We thank S. Milia and F. Abella from the the Experimental Histopathology Facility of Inserm/ UPS/ ENVT US006 CREFRE-Anexplo, Toulouse Purpan for technical assistance. We thank S. Allart and L. Lobjois from the cellular imaging facility of Infinity for their helpful input and attempts to stain the sensory neuronal network in the gut. We also thank A. Canivet of the Inserm U1220 - IRSD organoid core facility, Toulouse, France, for performing image analysis. We are grateful to C. Deraison and N. Vergnolle for providing financial and managerial support (e.g., through ERC-310973 PIPE to NV), and for critical reading of the manuscript.

## Author contributions

**Conceptualization:** Alexis Audibert, Nicolas Cenac, Gilles Dietrich, Chrystelle Bonnart, Nicolas Blanchard.

**Data curation:** Alexis Audibert, Xavier Mas-Orea, Nicolas Cenac.

**Formal analysis:** Alexis Audibert, Xavier Mas-Orea, Matteo Serino, Nicolas Cenac.

**Funding acquisition:** Nicolas Blanchard.

**Investigation:** Alexis Audibert, Xavier Mas-Orea, Léa Rey, Marcy Belloy, Emilie Bassot, Louise Battut, Matteo Serino, Nicolas Cenac, Chrystelle Bonnart.

**Methodology:** Alexis Audibert, Xavier Mas-Orea, Léa Rey, Marcy Belloy, Emilie Bassot, Louise Battut, Gilles Marodon, Frederick Masson, Matteo Serino, Nicolas Cenac, Chrystelle Bonnart.

**Project administration:** Chrystelle Bonnart, Nicolas Blanchard.

**Resources:** Gilles Marodon, Frederick Masson, Matteo Serino, Nicolas Cenac, Gilles Dietrich.

**Supervision:** Chrystelle Bonnart, Nicolas Blanchard.

**Visualization:** Alexis Audibert, Matteo Serino, Nicolas Cenac.

**Writing – original draft:** Alexis Audibert.

**Writing – review & editing:** Alexis Audibert, Frederick Masson, Matteo Serino, Nicolas Cenac, Gilles Dietrich, Chrystelle Bonnart, Nicolas Blanchard.



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
