## [Decision Letter · Decision Letter 0]

5 Jan 2025

PPATHOGENS-D-24-02202

Toxoplasma gondii chronic infection decreases visceral nociception through peripheral opioid receptor signaling

PLOS Pathogens

Dear Dr. Blanchard,

Thank you for submitting your manuscript to PLOS Pathogens. After careful consideration, we feel that it has merit but does not fully meet PLOS Pathogens's publication criteria as it currently stands. Therefore, we invite you to submit a revised version of the manuscript that addresses the points raised during the review process.

Please submit your revised manuscript within 60 days Mar 06 2025 11:59PM. If you will need more time than this to complete your revisions, please reply to this message or contact the journal office at plospathogens@plos.org. Please include the following items when submitting your revised manuscript:

We look forward to receiving your revised manuscript.

Kind regards,

Tracey J. Lamb

Section Editor

PLOS Pathogens

Tracey Lamb

Section Editor

PLOS Pathogens

 Sumita Bhaduri-McIntosh

Editor-in-Chief

PLOS Pathogens

orcid.org/0000-0003-2946-9497

 Michael Malim

Editor-in-Chief

PLOS Pathogens

orcid.org/0000-0002-7699-2064

**Journal Requirements:**

At this stage, the following Authors/Authors require contributions: Alexis Audibert, Xavier Mas-Orea, Léa Rey, Marcy Belloy, Emilie Bassot, Louise Battut, Gilles Marodon, Frederic Masson, Nicolas Cenac, Gilles Dietrich, Chrystelle Bonnart, and Nicolas Blanchard. Please ensure that the full contributions of each author are acknowledged in the "Add/Edit/Remove Authors" section of our submission form.

https://journals.plos.org/plospathogens/s/submission-guidelines#loc-parts-of-a-submission

5) We notice that your supplementary Figures are included in the manuscript file. Please remove them and upload them with the file type 'Supporting Information'. Please ensure that each Supporting Information file has a legend listed in the manuscript after the references list.

Potential Copyright Issues:

i) Figures 1A, 1F, 4A, S3A, and S3C. Please confirm whether you drew the images / clip-art within the figure panels by hand. If you did not draw the images, please provide (a) a link to the source of the images or icons and their license / terms of use; or (b) written permission from the copyright holder to publish the images or icons under our CC BY 4.0 license. Alternatively, you may replace the images with open source alternatives. See these open source resources you may use to replace images / clip-art:

7) We note that your Data Availability Statement is currently as follows: "All relevant data are within the manuscript and its Supporting Information files." Please confirm at this time whether or not your submission contains all raw data required to replicate the results of your study. Authors must share the “minimal data set” for their submission. PLOS defines the minimal data set to consist of the data required to replicate all study findings reported in the article, as well as related metadata and methods (https://journals.plos.org/plosone/s/data-availability#loc-minimal-data-set-definition).

8) Please amend your detailed Financial Disclosure statement. This is published with the article. It must therefore be completed in full sentences and contain the exact wording you wish to be published.

**Comments to the Authors:**

**Please note that one of the reviews is uploaded as an attachment.**

**Reviewers' Comments:**

Reviewer's Responses to Questions

**Part I - Summary**

Reviewer #1: The authors' work provides the detailed mapping of immune changes in the colon during latent infection and reveals a new consequence of T. gondii chronic infection: a decreased visceral nociceptive response to colorectal distension. Overall, the article is convincing, with solid experiments and high-quality writing. However, the following points need improvement before the manuscript can be considered for publication.

Reviewer #2: T. gondii is a food-borne parasite that commonly infects humans. Upon ingestion of oocysts or tissue cysts, sporozoites or bradyzoites are released, which convert into tachyzoites that pass the epithelial barrier. After reaching the lamina Propria, they disseminate throughout the body. Intestinal inflammation, mainly driven by Th1 cytokines, is observed with high doses of tissue cysts upon oral infection in the acute phase of infection in certain inbred mouse strains (BL/6 e.g.) and is primarily localized to the distal part of the small intestine. These lesions revert to normal during the chronic phase of infection and resemble those associated with inflammatory bowel disease (IBD, particularly Crohn´s disease).

In the present study by Audibert et al., the Blanchard lab addressed the question to what extend T. gondii promotes the development of PI-IBS (Irritable Bowel Syndrome, not to be confused with IBD) by influencing the neuroimmune environment of the gut. To this end, the effects of T. gondii on the microenvironment of the colon and on visceral nociceptive responses were investigated in a mouse model of chronic infection.

In general, the quality of the data is good, however, certain experiments should be repeated and/or more animals included. The decrease in visceral nociception in chronically infected mice is interesting but an explanation of the mechanism and cell types involved its missing. Without these data, the paper is rather descriptive and does not advance the field sufficiently.

After initially comparing i.p. and oral infections, the authors subsequently decided to use only tachyzoites for the remainder of the study, as this approach allows for a reduction in the number of donor animals required. This is certainly commendable; however, in my opinion, the decision to use the i.p. model instead of the oral model reduces the physiological relevance of the study. I suggest adding colon thickness and length in Figures 1E and 1F for the oral model of infection. Including this data would help to rule out macroscopic differences between both infection models and would give more weight to the decision to select the i.p. model of infection.

These are my two major concerns with the present study.

Reviewer #3: The manuscript by Audibert et al. sets out to determine if Toxoplasma infection could be a pathogen that triggers PI-IBS. Several gastrointestinal pathogens have been shown to play a role in instigating PI-IBS and because Toxo has been shown to cause substantial inflammation that is a kin to Crohn’s disease the authors set out to determine if it could also participate in IBS. However, these are two very different diseases with IBD being driven by inflammation directed at the microbiota and IBS having alterations in the function and pain response of the gut. The study uses the PruGRA6OVA model of Toxo which results in latency in B6 mice. The authors characterize the chronic immune environment in the colon of the mice infected by both IP and PO and find that little to no parasite exists during the chronic phase of infection. They also test the VSM to determine the nociceptor response in the colon and surprisingly find it reduced. They go on to determine if the Penk, a precursor to enkephalin, is involved in the dampening of the nociceptor pain response. They use a T cell specific Penk system to test if it is derived from T cells but it is not. It was difficult to understand the justification for the T cells specific KO of Penk. They also block all mu opioid receptor and show that rescues the pain response only in the infected animals. This study is of interest in understanding how chronic infection and inflammation can alter a tissue and its function long-term. There are strengths to the study, however it seems premature in some of the major conclusions, ie Toxo could be negatively associated with IBS and there are some experiments that could be done to deepen the understanding of how this system is being altered that the authors have proposed in the discussion but have not done yet. The study as it stands is of interest in how this infection impacts the pain response, however there is limited mechanistic data, and the data supporting Toxo and IBS connection seems premature, together this limits the enthusiasm for the overall findings as it stands.

**Part II – Major Issues: Key Experiments Required for Acceptance**

Reviewer #1: The authors suggest that "Toxoplasma gondii chronic infection decreases visceral nociceptive responses through peripheral opioid signaling." Given that opioids are analgesic molecules, it would be expected that administering antagonists to opioid receptors would exacerbate pain phenotypes in pain models, even in the absence of infection. Consequently, in Figures 4B and 4C, the comparison between only two groups was insufficient to conclusively demonstrate that the observed reduction in visceral pain during infection is mediated by opioid signaling. To strengthen this claim, the authors should consider including additional experimental groups and conducting further experiments.

Reviewer #2: Fig. 1B

The authors report how many animals were used in this study, but did all the animals survive the infection? If not, are there significant differences between the respective routes of infection? And were all the animals - especially those showing no weight loss - really infected? A conventional test for seroconversion would be necessary to confirm this.

Fig. 1C

Were the animals that were infected in 1B examined here? Does this mean that, in the case of the oral infection, 10 animals died spontaneously?

Fig. 1D

Here, data from three independent repetitions are pooled, with 15, 6, and 12 animals respectively. Are these the respective animal numbers per repetition, or the total numbers? If the former, please specify. If the latter, then significantly more animals need to be included per repetition (especially for the oral infections), also in order to make statistically robust conclusions.

Fig. 1E

Why are the pooled data from both (two) experiments not shown in this case?

As mentioned above, determining colon thickness and length following oral infection is necessary in order to (better) justify the subsequent approach (i.p. infection).

Fig. 4B

Again here, why are the pooled data from both (two) experiments not shown in this case?

Fig. S3

There seems to be an issue with the labeling of the individual panels here. Please correct accordingly.

This experiment should be repeated in order to ultimately present the pooled data (from at least two experiments).

Fig. S4

As an important control for Fig. 4, data from more than just a single experiment with 6 mice should be included here.

Reviewer #3: 1. The authors do not show the data that Penk is the only one detected or what tissues it is detected in, what is the level in the colon? In various cells from the colon? What about the brain? This data is stated in the discussion but not shown and is critical for understanding the expression and how infection alters this expression. The authors have the FlowFish worked out and showed it nicely in the KO mice.

The most compelling data is the opioid receptor inhibition but without supporting data to understand the source of Penk it is difficult to understand what is the source driving the Penk increase.

2.In the IP route what do the authors know of the contribution of the adrenal gland to the production of Penk/Enkephalins? This is also a major source in the body but is not considered in the IP infection model, again does Penk increase in the adrenal gland over the course of infection?

3 What is the percentage of the TRM in CD4 and CD8 that are specific to OVA and or Toxo? It is hard to interpret the conclusions of the gut as it is not clear the parasite specific TRM compared to the commensal/other TRM. Also, it is difficult to make broad conclusions from other studies on these due to the influence of vendor and animal facilities.

4. The discussion could use some more details (line 320-329ish) in the cells that express Penk and cells that express the receptors, these are referred to but it is not the easiest to follow.

5. The authors discuss the role of the microbiota in producing signals that can trigger neurotransmitters but do not investigate the dysbiosis in their model or if any of the bugs are in their model. Does antibiotic treatment alter the VSM response in infected mice? This is a straightforward experiment that could answer this question and provide more context into what the authors are studying.

6. Is serotonin altered in the gut during chronic infection? As this could also be participating and many stimuli are altered by Toxo infection it seems like this should be measured.

**Part III – Minor Issues: Editorial and Data Presentation Modifications**

Reviewer #1: During chronic infection, despite elevated enkephalins released by Trm, is there a statistically significant increase in the tissue content of enkephalins? Additionally, does the content of enkephalins in the tissue significantly decrease after the knockout of enkephalins secreted by Trm?

Would it be possible to measure Penk mRNA expression in other immune cells via flow cytometry, similar to what was done in Figure S3B? If other immune cells besides CD4+ T cells have the potential to release enkephalins, this would be valuable information.

Figure 1F does not indicate the number of mice used, each figure legend should clearly state that.

Reviewer #2: Fig. 2A

In all panels except the first, 12 points for infected animals are displayed, even though the legend states that only 11 animals were used. Please correct the Fig. accordingly.

Fig. 2C and D

The selection of FACS plots does not seem to effectively represent the data shown. Are there no plots that better exemplify the data presented?

Fig. 3A and B

As in Fig. 2A, 12 T. gondii infected animals are shown here, although the legend states 11. Please correct this.

Reviewer #3: (No Response)

PLOS authors have the option to publish the peer review history of their article (what does this mean? ). If published, this will include your full peer review and any attached files.

**Do you want your identity to be public for this peer review?** For information about this choice, including consent withdrawal, please see our Privacy Policy .

Reviewer #1: **Yes: ** Yan Li

Reviewer #2: No

Reviewer #3: No

**Figure resubmission:**
---

## [Decision Letter · Decision Letter 1]

7 Apr 2025

Dear Dr Blanchard,

We are pleased to inform you that your manuscript 'Toxoplasma gondii chronic infection decreases visceral nociception through peripheral opioid receptor signaling' has been provisionally accepted for publication in PLOS Pathogens.

Best regards,

Tracey J. Lamb

Section Editor

PLOS Pathogens

Tracey Lamb

Section Editor

PLOS Pathogens

Sumita Bhaduri-McIntosh

Editor-in-Chief

PLOS Pathogens

orcid.org/0000-0003-2946-9497

Michael Malim

Editor-in-Chief

PLOS Pathogens

orcid.org/0000-0002-7699-2064

Reviewer Comments (if any, and for reference):

Reviewer's Responses to Questions

**Part I - Summary**

Reviewer #2: The authors have clearly invested considerable effort, resulting in a significantly improved manuscript. All of my concerns have been thoroughly addressed to my satisfaction. I just have a few comments. Although I would appreciate these being considered, they should not hinder my recommendation for publication.

Reviewer #3: The authors have done an excellent job of addressing all the concerns.

Reviewer #4: This study by Audibert and colleagues provides compelling evidence that chronic Toxoplasma gondii infection reduces visceral pain via opioid signaling. The experiments are robust, clearly presented, and the manuscript is well-written. After reviewing the iterations between other referees and the authors, I believe that most concerns have been adequately addressed. However, I agree that the mechanism underlying this intriguing phenomenon remains somewhat elusive. Since the other referees have covered many aspects of the manuscript, I will focus specifically on the mechanism, which still lacks a comprehensive explanation.

Although irritable bowel syndrome (IBS) was initially central to the framework of this study, the results do not strongly support a direct causal link between T. gondii infection and IBS. In light of this, I recommend refocusing the manuscript to emphasize the broader, fundamental host-pathogen interactions. T. gondii infection leads to significant rewiring of the brain to manipulate host behavior in ways that support the parasite’s lifecycle. The discovery that T. gondii induces hypoalgesia, whether centrally or peripherally mediated, represents an insightful contribution to understanding how this parasite affects its host. This finding adds to the growing body of knowledge about the complex strategies employed by T. gondii to hijack host behavior for its survival and transmission.

A particularly striking finding is the reduction in VMR to colorectal distension in both per os and ip-infected mice, regardless of intestinal parasite persistence in the former. While enkephalin production by T cells does not seem to drive this effect, the administration of naloxone methiodide reversed the hypoalgesic state. The authors suggest that peripheral opioid receptors may be involved, but the precise mechanism remains unclear. While additional efforts were made to explore the role of other opioids or mucosal cell-derived enkephalins, the results were inconclusive.

The study presents a novel and significant finding by demonstrating that T. gondii infection alters pain perception, with potential implications for understanding host-parasite interactions and pain regulation. The well-executed experimental design and clear presentation strengthen the impact of the study. However, despite these strengths, the limited mechanistic insight into the observed hypoalgesia represents a gap in the current understanding.

**Part II – Major Issues: Key Experiments Required for Acceptance**

Reviewer #2: (No Response)

Reviewer #3: (No Response)

Reviewer #4: Peripheral Nerve Recording - To validate the involvement of peripheral opioid receptors in the hypoalgesic state observed, I recommend performing peripheral nerve recordings from colonic sensory afferents in infected mice, along with naloxone methiodide or vehicle administration. This experiment could directly assess whether the depolarization threshold is altered in the peripheral nerve endings of infected mice, while also providing crucial evidence for peripheral opioid receptor involvement.

Evaluation of Potential Central Involvement - A somewhat provocative, but important, consideration is whether naloxone methiodide could be acting centrally in this model. While it is generally believed that naloxone methiodide does not readily cross the blood-brain barrier (BBB), it has been shown that at doses higher than 2 mg/kg, it may cross the BBB to a limited extent (PMID: 31964994). Since there were no changes in nociception in non-infected mice, BBB crossing is unlikely solely due to the dosage employed in this study. Nonetheless, there is a possibility that T. gondii infection might increase BBB permeability, thereby allowing naloxone methiodide to reach the brain at pharmacologically relevant concentrations. A study published in eLife has shown that although T. gondii invasion in the early stages does not cause significant BBB disruption, focal permeability increases were observed near parasite foci at later stages (PMID: 34877929). It would be useful to perform a simple behavioral test like the Tail Flick Test (or any other accepted method) to evaluate potential central opioid involvement.

**Part III – Minor Issues: Editorial and Data Presentation Modifications**

Reviewer #2: Fig. 1

If I understand correctly, Figure 1B shows the survival curve for all the mice used in the experiments of Figure 1. I cannot follow the rationale behind the decision for the animal numbers for the respective analyses. For example, 22 brain IP animals but only 10 colon IP. 19 brain PO but only 7 colon PO.

When I count the uninfected animals, I come up with 36, but the authors indicate that 37 were used (1B). There may be good reasons for this, but they are not clear to me. This could be presented more clearly.

Fig. 1D

I appreciate that the authors conducted an additional experiment. However, I still believe that 6 animals is too few to make robust statistical conclusions. Therefore, aside from the fact that no animal numbers are mentioned in panel A, I would strongly recommend integrating the new data into Figure 1D.

Fig. 2C/D

The authors have some variability – especially in panel C “T. gondii” – which is of course completely fine. The question was whether there might be a plot that better represents the observed strong differences. This was more of a comment than a requested change.

Reviewer #3: (No Response)

Reviewer #4: Nothing to highlight.

PLOS authors have the option to publish the peer review history of their article (what does this mean? ). If published, this will include your full peer review and any attached files.

**Do you want your identity to be public for this peer review?** For information about this choice, including consent withdrawal, please see our Privacy Policy .

Reviewer #2: No

Reviewer #3: No

Reviewer #4: No

---

## [Editor Report · Acceptance letter]

Dear Dr Blanchard,

We are delighted to inform you that your manuscript, "*Toxoplasma gondii* chronic infection decreases visceral nociception through peripheral opioid receptor signaling," has been formally accepted for publication in PLOS Pathogens.

Best regards,

Sumita Bhaduri-McIntosh

Editor-in-Chief

PLOS Pathogens

orcid.org/0000-0003-2946-9497

Michael Malim

Editor-in-Chief

PLOS Pathogens

orcid.org/0000-0002-7699-2064